# microRNAs as Biomarkers of Endothelial Dysfunction and Therapeutic Target in the Pathogenesis of Atrial Fibrillation

**DOI:** 10.3390/ijms24065307

**Published:** 2023-03-10

**Authors:** Vanessa Desantis, Maria Assunta Potenza, Luca Sgarra, Carmela Nacci, Antonietta Scaringella, Sebastiano Cicco, Antonio Giovanni Solimando, Angelo Vacca, Monica Montagnani

**Affiliations:** 1Department of Precision and Regenerative Medicine and Ionian Area, Pharmacology Section, University of Bari Aldo Moro Medical School, 70124 Bari, Italy; 2General Hospital “F. Miulli” Acquaviva delle Fonti, 70021 Bari, Italy; 3Department of Precision and Regenerative Medicine and Ionian Area, Unit of Internal Medicine and Clinical Oncology, University of Bari Aldo Moro Medical School, 70124 Bari, Italy

**Keywords:** atrial fibrillation, atrial fibrosis, endothelial dysfunction, microRNAs

## Abstract

The pathophysiology of atrial fibrillation (AF) may involve atrial fibrosis/remodeling and dysfunctional endothelial activities. Despite the currently available treatment approaches, the progression of AF, its recurrence rate, and the high mortality risk of related complications underlay the need for more advanced prognostic and therapeutic strategies. There is increasing attention on the molecular mechanisms controlling AF onset and progression points to the complex cell to cell interplay that triggers fibroblasts, immune cells and myofibroblasts, enhancing atrial fibrosis. In this scenario, endothelial cell dysfunction (ED) might play an unexpected but significant role. microRNAs (miRNAs) regulate gene expression at the post-transcriptional level. In the cardiovascular compartment, both free circulating and exosomal miRNAs entail the control of plaque formation, lipid metabolism, inflammation and angiogenesis, cardiomyocyte growth and contractility, and even the maintenance of cardiac rhythm. Abnormal miRNAs levels may indicate the activation state of circulating cells, and thus represent a specific read-out of cardiac tissue changes. Although several unresolved questions still limit their clinical use, the ease of accessibility in biofluids and their prognostic and diagnostic properties make them novel and attractive biomarker candidates in AF. This article summarizes the most recent features of AF associated with miRNAs and relates them to potentially underlying mechanisms.

## 1. Introduction

Atrial fibrillation (AF), one of the most common cardiac arrhythmias, is associated with a high risk of mortality and morbidity related to severe complications, including congestive heart failure, arterial embolism, myocardial infarction, and cerebral stroke [1].

Despite the more refined therapeutic approaches proposed in recent years, most currently available pharmacological therapies are still limited by an incomplete efficacy, a moderate risk of adverse reactions, and a significant long-term recurrence rate [2]. Moreover, none of the existing treatments (including both catheter ablation and drug treatments) seem able to prevent or halt arrhythmia progression to more therapy-resistant forms [3]. On these bases, a deeper understanding of the potential molecular mechanisms underlying the onset and the progression of AF may help to identify critical therapeutic targets and clinically relevant mechanisms, with potential translational significance on the clinical management of patients.

Increasing attention has been devoted to the molecular changes controlling atrial structural remodeling, prominently including tissue fibrosis, which might contribute to conduction abnormalities and become a risk factor for AF. In turn, AF may promote its own maintenance by concurring to atrial fibrosis (Afib) and remodeling. Indeed, the extent of Afib has been proposed as a predictor of AF-recurrence after catheter ablation [4].

In the complex cell to cell interplay that triggers fibroblasts (FBs) to differentiate into profibrotic collagen-secreting myofibroblasts, endothelial cell dysfunction (ED) might play an unexpected but significant role. Indeed, several possible mechanisms sustain the association between AF and (ED), including the impaired rheology after AF-induced turbulent flow, the reduced production of nitric oxide (NO) in the left atrium, the unbalanced release of endothelin-1 (ET-1), and the activation of systemic factors, such as the renin-angiotensin aldosterone system (RAAS) and inflammatory signaling that may sustain and promote ED progression. Whether ED is a biomarker of AF or an intermediate step in a causal pathway developing AF, the tight and bidirectional interrelation between these two conditions is unquestionable, potentially feeding a vicious cycle that may worsen ED and lead to persistent AF [5].

microRNAs (miRNAs) are small, single-stranded RNA molecules of about 22 nucleotides in length; miRNAs regulate gene expression at the post-transcriptional level by binding to specific sites within the mRNA target, at the 3′ untranslated region (3′-UTR), in order to determine either activation or inhibition [6]. miRNA regulation involves a broad range of biological processes, such as cell differentiation, proliferation, and apoptosis [7]. Aberrant miRNAs expression is associated with various cardiovascular diseases [8]. Interestingly, an essential role of miRNAs has recently been hypothesized in the control of plaque formation, lipid metabolism and angiogenesis, in cardiomyocyte growth and contractility, and even in the development and maintenance of cardiac rhythm [9].

miRNAs may be contained in exosomes, extracellular vesicles carrying a variety of biological substances [10] able to modulate basic cellular functions [11,12]. The emerging role of exosomal miRNAs in cardiac regeneration and protection points out their importance in a direct cell-to-cell interaction and suggests their potential use in diagnosing and treating AF promoting conditions [13]. In this respect, the advent of new therapeutic approaches improving the exosome stability and modifying surface epitopes might provide new specific pharmaceutical agents to target cells and tissues in vivo. Moreover, since miRNAs are released within the body fluids (i.e., peripheral blood [PB], serum and plasma), they could be attractive as non-invasive biomarkers to monitor cardiovascular diseases and, as a future direction, possible therapeutic approaches for AF treatments.

## 2. Molecular and Cellular Mechanisms Linking Atrial Remodeling and Endothelial Dysfunction to AF

The molecular and cellular mechanisms underlying AF initiation include atrial ectopic triggers and factors facilitating a reentry-prone substrate. Electrophysiological changes characterizing AF usually initiate with a rapid atrial activation, known as “triggered activity”, which may then perpetuate via a reentry pattern where the abnormal impulse propagates through a circuit in a fast and repetitive way [14,15].

The spontaneous atrial activity in AF onset has mainly been linked to ion channel dysfunctions [16,17], inappropriate autonomic tone, and abnormal Ca^2+^ handling [18,19].

The dysregulation of both inflammatory and immune pathways may significantly contribute to ion channel dysfunction and AF initiation on one side, and unrestrained FBs proliferation and atrial remodeling on the other [13]. In turn, the interaction between FBs and atrial myocytes, crucial in the Afib process [13], may enhance atrial electrical and functional remodeling, which is responsible for the progression and self-maintenance of AF [20]. 

Afib accounts for the inhomogeneity of conduction and tortuosity of impulse propagation [21]. Interstitial fibrosis is associated with impaired atrial function in patients with AF [22], and the content and distribution of atrial fibrous tissue seems to correlate directly with the risk for complications and therapeutic failure in AF patients [23]. Afib is characterized by excessive extracellular matrix components (ECM) deposition from cardiac FBs and myofibroblasts. Under normal conditions, resident FBs are small spindle-shaped and quiescent cells that control the ECM composition and structure [24]. Profibrotic stimuli cause the proliferation and differentiation of FBs into myofibroblasts that increase collagen production [25]. Myofibroblasts acquire contractile properties, including the expression of α-smooth muscle actin (α-SMA) that represents a hallmark of FB transition to new myofibroblast phenotype and contributes to pathologic cardiac remodeling [4].

The source of activated cardiac FBs that accumulate in response to pathological insults remains unclear. However, resident FBs, endothelial cells (ECs), bone-marrow-derived progenitor cells, and perivascular cells have been proposed among precursors of myofibroblasts [26]. During embryonic development, under the influence of several growth factors, the primary pool of resident, quiescent cardiac FBs originate from the mesenchyme via epithelial-mesenchymal transition (EpiMT) and endothelial to mesenchymal transition (EndMT), taking part to ECM homeostasis [25]. Under pathological conditions, a secondary pool of FB-like cells originating from various cellular sources, including epithelial and endothelial cells via EMT and EndMT, contributes to fibrogenesis and cardiac remodeling. The fibrotic process stimulates the proliferation of FBs and their differentiation into myofibroblasts, with an increasing ability to synthesize ECM proteins, including fibronectin, procollagen (later converted in mature collagen), collagen cross-linking agents and enzymes, such as matrix metalloproteinases (MMPs), that modify the ECM [13,25].

### 2.1. Profibrotic Stimuli Involved in Atrial Structure Remodeling Related to AF

Atrial structure remodeling is associated with several profibrotic stimuli, such as angiotensin II (Ang II), transforming growth factor-β (TGF-β), platelet-derived growth factor (PDGF), oxidative stress and inflammation, which are all involved in the excessive deposition of ECM proteins by FBs that impairs mechano-electric coupling of cardiomyocyte and increases the risk of arrhythmogenesis and mortality [26].

*Sympathetic receptor activation*—Autonomic imbalance is one of the most important pro-arrhythmic modulating factors, favoring both atrial ectopic activity and reentry. Sympathetic stimulation of adrenergic receptors enhances all processes controlling Ca^2+^ entry, storage, and release in cardiomyocytes, promoting significant and heterogeneous changes of atrial electrophysiology that may represent both a substrate and a trigger for the onset of AF. Downstream β1-adrenergic receptors, activation of PKA-dependent phosphorylation of several Ca^2+^-handling proteins, and cyclic adenosine monophosphate (cAMP) production [27] results in increased sarcoplasmic-reticulum (SR) Ca^2+^ load. This effect, together with hyperphosphorylation of cardiac ryanodine-receptor channel type 2 (RYR2), can cause diastolic SR Ca^2+^ leak, promoting delayed afterdepolarization (DAD), which is a recognized source of ectopic activity in the onset and maintenance of AF [27,28]. In addition to β1-adrenoreceptors, stimulation of α1-receptor-mediated signaling may also contribute to the SR Ca^2+^ leak in atrial cardiomyocytes, as well as to the maintenance of AF [29]. Activation of α1-adrenergic receptors can also inhibit inward rectifying K^+^ current (IK), which is important in setting the resting potential and the repolarization reserve. By inhibiting IK, α1-adrenergic activation may enhance stimulation of cardiac neurons and increase automaticity, promoting AF onset [15,29]. Thus, sympathetic receptor activation may induce atrial ectopic activity via multiple mechanisms, contributing to the pathogenesis of atrial arrhythmias, which in turn exacerbates atrial autonomic imbalance.

*Ca^2+^ influx*—FB function is regulated by Ca^2+^ influx, whose intracellular availability depends on several pathways, including the activation of receptor-operated channels (ROC: responsive to agonists such as angiotensin II), store-operated calcium channels (SOCC: opened in response to endoplasmic reticulum Ca^2+^ depletion), and transient receptor potential channels (TRP), promoting FBs differentiation into collagen-secreting myofibroblasts [4]. The driving force for Ca^2+^ entry into the FBs depends on the potential resting membrane, which in turn is controlled by the inward rectifier K^+^ channel (I k1) [17]. As ion channels of atrial FBs appear to play a significant role in the control of FB activation and in the development of fibrosis, targeting Ca^2+^ handling is a therapeutic option. Indeed, a direct electrotonic coupling between myofibroblasts and cardiomyocytes has been suggested to cause cardiomyocyte depolarization and promote triggering activity [30,31].

*Angiotensin II (AngII)*—Ang II represents the predominant effector of RAAS. Ang II linking to Ang II type 1 receptor (AT1R) induces cell proliferation, migration, and protein synthesis in cardiac FBs [32], and results in rapid pacing-induced AF in experimental animal studies [33]. Furthermore, as Ang II is expressed and activated by macrophages and myofibroblasts, it is also involved in the local inflammatory response [34]. Ang II activates NADPH oxidases, increasing reactive oxygen species (ROS)-mediated stimulation of downstream effectors. Excessive ROS production and subsequent oxidative stress may contribute to abnormal Ca^2+^ -handling and Ca^2+^ -overload, which triggers Ca^2+^-dependent signaling and atrium hypocontractility. In particular, Ca^2+^ -induced activation of the nuclear factor of activated T cells (NFAT), that inhibits gene expression of both L-type calcium channel subunits and transcription of miRNA-26, may contribute to the downregulation of L-type Ca^2+^ -current and activation of Kir2.1 and TRPC3 channels. These genetic modifications decrease action potential duration, causing AF-induced electrical remodeling [35]. Moreover, Ang II increases the production of TGF-β and ECM proteins that form the fibrous tissue, promoting AF-induced fibrotic remodeling [36]. Collagen deposition induced by Ang II in cardiac FBs requires TGF-β/Smad and mitogen-activated protein kinase (MAPK) signaling [34,37].

*Transforming Growth Factor β1 (TGF-β1)*—Among the components of the TGF-β family, TGF-β1 is particularly important as a mediator of FBs activation. The canonical pathway of TGF-β1 involves the Smad2/3 phosphorylation and the subsequent binding to Smad4, whose nuclear translocation activates numerous profibrotic genes. In addition, TGF-β occurs to modulate FB phenotype and gene expression, and promotes ECM deposition by upregulating collagen and fibronectin synthesis and by decreasing matrix degradation via protease inhibitors induction [38]. Overexpression of TGF-β1 in mice causes atrial fibrosis and increases AF susceptibility [39]. Interestingly, both TGF-β1 and Ang II-mediated FBs activation is higher in atrial than in ventricular tissue [40]. A recent investigation shows that TGF-β1 stimulation of human endocardial ECs is able to induce EndMT by upregulation of miR-181B expression [41]. Although further validation is needed, these findings are consistent with the recent observation that EndMt is implicated in fibrotic atrial remodeling during the development and progression of human AF [42].

*Oxidative stress—*Experimental and clinical approaches suggest that NADPH oxidase 2 (NOX2) and NOX4 are significant sources of ROS production in AF. Ang II, TGF-β, and atrial stretch-mediated activation of these enzymes enhance the ROS-generating systems, resulting in sustained oxidative stress, which stimulates myocyte apoptosis, atrial inflammation, fibrosis, and structural and electrical remodeling promoting AF [43]. Increased levels of ROS, such as superoxide (O_2_-) and hydrogen peroxide (H_2_O_2_), have been found to be associated with AF and are often accompanied by a decreased NO availability. Hydroxyl radical (OH-) and peroxynitrite (ONOO-) mediate the oxidative damage of myofibrils in AF, which in turn contribute to structural atrial remodeling [44]. Moreover, oxidative stress-induced mitochondrial DNA damage in AF causes Ca^2+^ overload by modulating Ca^2+^ handling proteins or channels, which can promote atrial electrical remodeling [45]. Endocardial NO deficiency, due to ROS production and eNOS uncoupling, may also contribute to thromboembolic complications observed in AF [43].

*Inflammation—*As a major contributor to the formation of fibrous tissue and atrial remodeling, and a significant player in ED, platelet activation, and coagulation cascade activation, inflammation may support both onset/maintenance of AF and its thromboembolic complications [46]. Several inflammatory cytokines, such as interleukin 6 (IL-6), tumor necrosis factor-α (TNF-α), IL-2, and IL-1β, are essential in promoting fibrosis. By acting as a transcription factor for both inflammatory cytokines and ROS, NFkB further exacerbates inflammatory processes [47]. TNF-α may also subsidize Afib by regulating MMPs activity and ECM proteins degradation [48]. A significant, positive correlation is found between NFkB activity, serum TNF-α, and IL-6 levels and collagen volume fraction in atrial tissue from AF patients with valvular heart disease [49]. On the same line, serum levels of fibro-inflammatory biomarkers, including MMP-9, type III collagen, and C-reactive protein (CRP), are higher in persistent-AF patients compared to control patients with sinus rhythm, and are positively correlated with echocardiographic left atrial volume, an index of atrial remodeling [50]. The presence of inflammatory macrophage markers further supports the close association between inflammation and fibrosis colocalized in atrial subendocardial tissue of patients with AF [51]. Leukocyte infiltration has been observed in atrial tissues of AF subjects [52]. Similarly, higher plasma concentrations and atrial deposition of oxidizing myeloperoxidase (MPO), secreted from infiltrated leukocytes, were found in AF patients when compared to control patients with sinus rhythm [53]. Interestingly, exacerbation of atrial fibrosis, increased MMP, and AF-promotion activity following chronic Ang II infusion was attenuated in MPO-KO mice [53]. Despite undoubtful evidence of the close link between inflammation and AF, whether inflammation is the cause or the consequence, and which specific inflammatory mediator may increase the susceptibility of the atria to fibrillation, remains to be clarified.

### 2.2. Changes in Endothelial Function and Correlation with AF

ED is strictly connected to cardiovascular pathology, being recognized as the earliest event in vascular dysregulation that, in turn, promotes vessel atheromasia, a pro-thrombotic milieu and cardiac arrhythmias [54]. 

The most crucial endothelium-derived compound is NO, a gaseous mediator directly involved in vasodilation, inhibition of platelet activation, muscular cell proliferation and migration, and inhibition of surface adhesion molecule expression, including intercellular adhesion molecule-1 (ICAM1) and vascular cell adhesion molecule-1 (VCAM1) [55]. NO is produced by the conversion of L-arginine (L-arg) to citrulline via oxidation of NG-hydroxyl-l-arginine by the endothelial NO synthase (eNOS) enzyme [55]. During the early phase of endothelial activation/dysfunction, the impaired activity of the L-arg/eNOS synthase pathway results in reduced levels of NO and a concomitant increased generation of ROS. ROS, in turn, limits NO production even more through peroxynitrite formation, promoting vascular leukocyte adhesion and contributing to the inflammatory *milieu* in the vasculature [56]. Moreover, the unbalanced NO/ROS production, and the resulting oxidative stress, hinders angiogenic and neo-angiogenic processes by affecting the endothelial progenitor cells (EPCs) mobilization, recruitment, functioning, and differentiation [57]. Thus, activation/dysfunction of the endothelium evolves toward an anatomical milieu subversion that disrupts a variety of distinct cell phenotypes (not excluding immune cells and FBs) and may significantly contribute to the clinical onset of AF and related cerebro-cardiovascular events [56]. In turn, AF-induced hemodynamic and structural changes may contribute to systemic and local ED [58].

Several mechanisms underlying AF pathogenesis and Afib progression, including oxidative stress, pro-inflammatory signaling, genetic renin-angiotensin axis abnormalities, and intracellular Ca^2+^ overload, are recognized actors of ED development. Activation of RAAS [59], neurohumoral activation [60], and oxidative stress directly impact both ED and AF. This is not unexpected since Ang II-mediated activation of the sympathetic system and increased ROS production are recognized triggers of ED and atrial fibrosis and remodeling [61]. Increased CRP and cytokine levels under AF-dependent atrial inflammation impact ECs as well [1]. 

Some genetic variants of key endothelium components, such as eNOS and caveolin-1, may help to explain the link between vascular disease and AF in young patients [52]. Similarly, genetic predisposition to abnormal RAAS function has been associated with AF [62] and vascular inflammation [63]. 

Additional factors, such as those described below, may contribute to impairing EC function under AF, acting in a reciprocal and vicious relationship that sustains and worsens both conditions. 

*Loss of shear stress and impaired NO availability*—Shear stress results from increased blood flow in the vessel. Loss of shear stress, as it occurs in turbulent flow conditions, reduces eNOS expression and function. Since AF predisposes to low and turbulent blood flow in the left atrium, AF is associated with a marked decrease in eNOS expression and NO bioavailability [64]. Moreover, since the left atrium is an endocrine organ releasing NO to systemic vessels, a disorganized atrial contraction, damaging the left atrial endocardium, may contribute to impairing systemic NO bioavailability [65]. This is in line with the observations of reduced plasma nitrite/nitrate levels in patients with AF [66]. Moreover, in left atria of pigs with induced AF, the impaired endocardial eNOS expression is coupled with increased plasminogen activator inhibitor-1 (PAI-1) expression [64]. Similarly, in rapid atrial pacing performed on rabbits, a decreased shear stress, caused by a fast and irregular pulse, impairs eNOS activities and increases oxidative stress markers [67]. 

Dimethylarginines, including asymmetric dimethylarginine (ADMA) and symmetric dimethylarginine (SDMA), are endogenous methylated analogues of L-arg, the precursor of NO. By competing with L-arg, ADMA can inhibit eNOS and contribute to impaired endothelial-mediated activities, oxidative stress, and inflammation in cardiovascular diseases [68,69,70]. SDMA does not inhibit eNOS directly but could interfere with the cellular uptake of L-arg, resulting in similar NO-impaired production. Elevated levels of ADMA and SDMA have been found in patients with AF, suggesting that an additional mechanism is shared between ED and AF [71,72,73,74].

*Increased ET-1 availability* –ET-1 is a peptide consisting of 21 amino acids produced by EC and smooth muscle cells (SMCs). It acts predominantly as a vasoconstriction mediator, but it is also involved in inflammation, cell adhesion, fibrosis, and angiogenesis [75]. ETA and ETB are the principal G-protein-coupled receptors that mediate the endothelin’s effects. ET-1 activities in vasal tissue are important to maintain vascular tone: ETA on SMCs mediate contraction, promoting the production of inositol 1,4,5-trisphosphate (IP3), and ETB balances this effect on ECs through the production of NO [75]. Thus, under conditions altering this balance, the vasoconstriction mediated by ET-1 prevails. Interestingly, ET-1 can stimulate the synthesis of collagen, the expression of ICAM-1, and the trans-differentiation of FBs into myofibroblasts [76,77]. Moreover, ET-1 seems to be involved in the EndoMT [78], where ECs change their phenotype into a mesenchymal and myofibroblastic phenotype, dissociate from the cell monolayer at the vessel lumen, and migrate toward the inner wall. During this migration, they lose specific endothelial markers (CD31, CD34, and VE-cadherin) and express mesenchymal or myofibroblastic markers like α-SMA and vimentin. As mentioned above, under atrial remodeling, the transformation of myocardial tissue into fibrotic tissue mainly results from the transition of FBs and ECs into myofibroblasts, leading to an increased production of collagen and EMC, and is accompanied by the stimulation of monocytes/macrophages, with the subsequent activation of inflammation, necrosis, and genesis of reparative fibrous tissue [79].

Thus, the inappropriate release of ET-1 in ED may represent an additional factor directly contributing to the morphological atrial changes underlying AF pathophysiology.

## 3. miRNAs Involved in Afib, ED, and AF and Their Putative Role as Biomarker

Expression levels of miRNAs may be a sign of the activation state of circulating cells, thus suggesting their importance as specific read-out of cell activation and tissue damage in response to cardiovascular risk factors [80,81]. Moreover, circulating miRNA levels may retain information on the miRNA expression changes in cardiac tissue and their involvement in myocardial remodeling, therefore providing new insights into the mechanisms of cardiovascular diseases [82]. Most circulating miRNAs can be easily identified with high specificity and sensitivity in the serum and plasma [80,83]. While the RNAs could degrade the exogenous miRNAs in plasma, the miRNAs associated with high-density lipoprotein (HDL) or included within extracellular vesicles, exosomes, and apoptotic bodies may become very resistant to RNase activity [8]. The ease of accessibility in biofluids and both the prognostic and diagnostic properties of circulating miRNAs make them novel and attractive candidates for biomarker development in cardiovascular diseases [83,84,85,86]. Current understanding on the role of down- and/or up-regulation of miRNAs support their contribution to the pathophysiology of AF, both in the regulation of atrial electrical/structural remodeling and in ED onset and progression [8,87]. The most relevant miRNAs associated with Afib, inflammation, and endothelial activities, and potentially involved in AF, are depicted in Figure 1.

### 3.1. Circulating miRNAs Associated with Electric and Atrial Remodeling

Experimental and clinical studies suggest that miRNAs-dependent gene regulation may contribute significantly to the initiation and maintenance of AF [82,88]. 

Among the miRNAs involved in AF electrical mechanisms, the increased expression of **miR-26** has been associated with a dysregulation in pro-fibrillatory inward-rectifier potassium current changes in AF murine model [89]. 

**miR-21** participates in the downregulation of Smad7 that reinforces the TGF-β1/Smad signaling pathway, thus promoting the development of Afib in AF, both in experimental study models and patients with aortic stenosis [88,90,91]. Effectors of TGF-β signaling were found to regulate miRNA-21 precursor transcription and its post-transcriptional maturation by overexpression of the DROSHA complex. Once activated, miR-21 negatively modulates several myocardial (SPROUTY1, PTEN, PDCD4) and extra-cardiac (RECK and TIP3) factors that play key roles in ECM homeostasis. The increased expression of miR-21 indirectly controls Afib via downregulation of SPRY1, a negative regulator of ERK-MAPK pathway, activating and promoting FBs proliferation and fibrosis [92]. Moreover, the miR-21-dependent inhibition of PDCD4 translation, resulting in anti-apoptotic effect in myocardiocytes, enhances the trans-differentiation of FBs into myofibroblasts and contributes to the reduction of MMPs expression [93,94,95]. 

In the process of cardiac remodeling, **miR-29a** is associated with both hypertrophy and fibrosis, representing a potential biomarker for myocardial remodeling assessment [96]. In response to pressure overload, the activation of TGF-β down-regulates miRNA-29, which in turn acts as a negative modulator of several mRNAs involved in fibrotic proteins synthesis, such as collagen, elastin, and fibrillin [96]. 

Four additional miRNAs dysregulated in AF patients (**miR-430-3p, miR-146b-5p, miR-630 and miR-367**) may influence mTOR and Hippo signaling pathways and have been indicated as prospective feasible biomarkers for the diagnosis of AF progression [97].

Other miRNAs differentially expressed in both human induced pluripotent stem cells (hiPSCs) and atrial tissue samples from AF patients have been recently evaluated [98]. In particular, **miR-34a** regulates atrial TASK-1 (tandem of P-domains in a weak inward rectifying K^+^ channel- related acid sensitive K^+^ channel 1) potassium channels and has been proposed as an important contributor to the initiation and maintenance of AF, linked to myocardial dilation. The correlation between clinical parameters and miRNA levels implicates a potential use of miRNAs as biomarkers of atrial cardiomyopathy. Analysis of miRNA expression might help to differentiate subtypes of AF [98].

### 3.2. Circulating miRNAs Associated with Vascular Inflammation and Endothelial Impairment

The pathophysiology of AF is characterized by high levels of circulating inflammatory markers [8,36]. For several of these inflammatory mediators, a correlation might be found with circulating miRNAs with a specific role in controlling endothelial function, vasculogenesis, neoangiogenesis, and SMCs proliferation and maturation.

**miR-10a** regulates the pro-inflammatory phenotype of ECs in atherosusceptible regions by inhibiting adhesion molecules, such as VCAM-1 and E-selectin, and controlling the NF-kB-mediated activity. The reduced expression of miR-10a is associated with enhanced synthesis of pro-inflammatory mediators as monocyte chemoattractant protein-1 (MCP-1), IL-6, IL-8, VCAM-1, and E-selectin. Similarly, **miR-181, miR-31,** and **miR-17-3p** are involved in controlling vascular inflammation [9].

**miR-Let-7g** maintains physiological endothelial functions by decreasing monocyte adhesion, inflammation, and senescence, and by increasing angiogenesis targeting genes of TGF-β and Sirt-1 signaling pathways. Loss of function in Let-7g is responsible for endothelial activation and subsequent vascular lesions. Moreover, because reduced plasma levels of Let-7g have been associated with higher circulating levels of PAI-1 in patients with lacunar stroke, Let-7g has been proposed as a risk factor for cardiovascular diseases [99].

In addition to its role in atrial remodeling, **miR-21** is also implicated in the modulation of the endothelial response to hemodynamic stress. The increase in miR-21 correlates with a reduced expression of the tumor suppressor PTEN, a negative regulator of the AKT/PKB signaling pathway promoting the EpiMT, EndoMT processes and myofibroblasts formation. PTEN also modulates the anti-apoptotic protein Bcl-2, which inhibits ECs apoptosis by promoting eNOS phosphorylation and NO release [9,95,100].

**miR-126**, which targets VCAM-1 and proinflammatory mediators, is considered the master regulator of endothelial homeostasis and vascular integrity. miR-126 exhibits indirect antithrombotic properties and attenuates vascular inflammatory responses (expression of VCAM-1 and fibrinogen, leukocyte counts) by targeting tissue factors in monocytes [101]. Its activities on vascular remodeling and decreased fibrosis are mediated via stimulation of hypoxia-inducible factor-1 (HIF-1), whose reparative effects are essential in protection against age-related vascular disease [102]. miR-126 expression is upregulated by the E26 transformation-specific (ETS)-1 transcription factor, which is induced by pro-inflammatory agents (e.g., TNF-α). This observation supports a negative feedback loop in which ETS-1-induced miR-126 expression counteracts TNF-α -mediated VCAM-1 synthesis [103]. Accordingly, conditional-knockout ECs miR-126 (miR-126EC-/-) mice exhibit significantly decreased cardiac function and increased cardiomyocyte hypertrophy, fibrosis, and inflammatory factor expression [104].

**miR-155** is one of the most extensively studied inflammation-associated miRNAs, which is a direct regulator of eNOS expression. It modulates TNF-α-induced suppression of eNOS level, decreasing endothelium-dependent vasorelaxation [105]. Moreover, the miR-155 family is involved in atherogenesis by promoting inflammatory response, degradation of lipoproteins and phagocyte cell debris, induced by cholesterol-loaded macrophages [106].

Higher levels of **miR-92a, miR-122, and miR-486** have been observed in patients with coronary artery disease compared with a control population [107]. Among them, **miR-122**, widely expressed in ECs, has been proposed as a potential biomarker for ED-associated atherosclerosis, heart failure, myocardial infarction (MI), and AF [108,109]. In experimental AF mice models, transfection with miR-122 inhibitor reduces cardiomyocytes apoptosis rate, upregulating the expression of anti-apoptotic protein Bcl-x and downregulating the pro-apoptotic protein caspase-3 and the phosphorylation levels of ERK1/2 [110,111].

**miR-223** is implicated in vascular ED [112], and its overexpression has been observed in plasma-derived exosomes of hyperlipidemic rats treated with paeonol, a potential anti-atherosclerosis molecule inhibiting the NLRP3 inflammasome pathway [112]. 

**miR-197** targets IL-6, ICAM-1, and regulates focal adhesion and vascular endothelial growth factor A (VEGFA) -receptor 2 (VEGFA-VEGFR2) pathways [113]. 

The circulating levels of **miR-125a-5p** and **miR-146a** appear up-regulated in sera from hyperlipidemic and/or hyperglycemic patients [114]. Similarly, **miR-1, miR-133a, miR-208a, miR-208b,** and **miR-499** are more expressed in patients with both coronary heart disease and heart failure and are proposed as biomarkers to assess the severity of myocardial damage [115].

### 3.3. Exosomal miRNA as Emerging Diagnostic Biomarkers in AF

As for circulating miRNAs, exosomal miRNAs may be involved in AF pathophysiology and development by affecting different biological mechanisms, including electric remodeling [116,117], structural remodeling [13,118,119], and energy metabolism processes [109,120,121].

The growing interest in exosomal miRNAs is due to their resistance to RNase activity, making these mediators more stable and potentially more suitable than circulating miRNAs as emerging diagnostic biomarkers in AF [13]. Importantly, **miR-142-5p, miR-483-5p, miR-223-5p,** and **miR-223-3p**, discovered in serum exosomes from patients with persistent AF, may be associated with the progression of AF [117]. The differential expression of exosomal **miR-92b-3p, miR 1306-5p,** and **Let-7b-3p**, in AF patients versus normal sinus rhythm (SR) patients, could be helpful for the characterization of the risk and progression of AF, therefore representing predictive biomarkers and novel therapeutic targets for AF [122].

Three additional exosomal miRs (**miR-382-3p, miR-450a-2-3p,** and **miR-3126-5p**), recently detected by bioinformatics analysis in human pericardial fluid, may also be representative of AF progression associated with cardiac fibrosis [109]. 

## 4. Could miRNA Modulation Become a Biomarker Guide in AF Therapy?

In heart failure and myocardial infarction, biomarkers are well established, and the major guidelines broadly refer to them for the management and diagnosis of these cardiovascular diseases [123,124,125]. Circulating troponins and natriuretic peptides, considered reliable biomarkers in other cardiovascular diseases, are suggestive of stroke and bleeding risk assessment in AF patients [126]. Moreover, the combination of blood-based biomarkers, recently evaluated in high-risk asymptomatic and cryptogenic stroke patients, has been proposed as applicable in AF screening strategies [127]. However, there are currently no suitable biomarkers for an early diagnosis and progression of this sustained arrhythmia [127,128]. Thus, the interest in identifying accurate and predictive biomarkers in AF management is actively pursued. 

### 4.1. miRNAs as Innovative Molecular Target Treatment in AF—Current Limits and Concerns

Molecular target treatment is the new frontier and is the promising “bullet” to treat different diseases [129]. Since gene target is one of the main focuses, targeting genetic alterations to customize AF treatment is an intriguing approach. The advantages of gene therapy for AF include tissue specificity, with less off-target effects, and increasing therapeutic effectiveness. Although genetic therapy may be designed for each patient based on specific disease characteristics, many challenges and unresolved issues still limit this potential novel approach. The inherent safety concern of using gene therapy to modify the myocardium is of paramount importance for clinical practice. 

Another point is how the genetic material would be delivered. The heterogeneity of the AF substrate may decrease efficacy in a single genetic alteration approach. Thus, a possible multi-gene analysis may help to identify specific targets [130].

On this general background, modulation of miRNAs expression, in vivo, is particularly interesting, despite several unresolved difficulties still halting this approach. First of all, a different strategic approach should consider modulating up-regulated versus down-regulated miRNAs. For down-regulated miRNAs, miRNA mimics may be the answer [131]. Mimics are synthetic double-stranded RNAs that are incorporated and processed by the cell-like endogenous miRNAs and, therefore, ‘mimic’ their effects [132]. The main issue on mimics is that they are not tissue- or cell-type specific, which may create undesirable off-target effects. This might be avoided by using cardiotropic adeno-associated virus-mediated miRNA transfer, which was already investigated on heart failure in mice [133] and cardiac hypertrophy in rats [134]. 

On the other hand, miRNAs over-expression can be suppressed by antago-miRNAs, locked nucleic acids, miRNA-sponges, or miRNA-masks. Antago-miRNAs are synthetic oligonucleotides with miRNA complementary sequences that bind to endogenous miRNAs, competitively inhibiting their binding to target genes [135], miRNA sponges [136], and erasers [135], and are multiple miRNA sequences incorporated into a vector. While sponges contain only the seed sequence and might inhibit various miRNAs, erasers are complement-specific miRNAs. miRNA-masks, instead, are single-stranded oligonucleotides that are complementary to a miRNA target sequence and can specifically block single miRNA–mRNA interactions.

As an additional general consideration, it should be kept in mind that a single miRNA treatment is rarely the best target. In AF pathophysiology, a significant difficulty is the large number of potential miRNAs that may be involved. Using microarray approaches, 285 miRNAs are up- and/or down-regulated in AF all together [28,89,90,92,137,138,139,140,141,142,143,144,145,146,147,148,149]. Among these, only 69 potential targets were validated using qRT-PCR, and alterations of only 9 of those miRNAs were associated with a specific dysregulation of proteins in AF-associated remodeling [131]. Specific data in animal models showed, for only 6 miRNAs (**miR-1, miR-21, miR-26, miR-29b, miR-31,** and **miR-328**), that normalizing their expression profiles may protect from AF induction or maintenance, emphasizing these miRNAs as promising therapeutic targets [89,92,137,142,147,149]. Although generally well preserved, miRNA expression levels may differ significantly between species. **miR-208a** isoform is predominantly expressed in rodents, whereas **miR-208b** is more prevalent in larger mammals. In addition, many clinical conditions may influence miRNA expression, making it particularly difficult, at this time, to establish which miRNA may display a decision-making role in the development of disease. Hopefully, among the many miRNA candidates as biomarkers, the ongoing research will help to clarify their specific activities in modulating fundamental processes underlying AF. For example, diabetes changes the miRNAs expression and their role in Afib; in a mice model, under hyperglycemic conditions**, miR-21-3p** acts as an inhibitor of adipose browning and participates in the process of Afib [150]. Therefore, downregulation of **miR-21-3p** may result in increased fibroblast growth factor receptor (FGFR)-1 expression, contributing to the activation of the browning transcriptional program via the FGFR1/FGF21/PPARγ pathway involved in Afib and/or AF. Nonetheless, results on modulation of AF-related miRNAs in animals may not be sufficient to evaluate the potential as a therapeutic target against AF in humans. Indeed, downregulation of **miR-26a** occurs consistently in dogs in response to pacing-induced AF [146] but changes in human AF are inconsistent [143,144,145]. As mentioned, miRNAs altered in AF are large in number, and multiple miRNAs are involved in controlling arrhythmogenicity in the heart. Although results from experimental studies in animal models suggest that restoring expression of one single miRNA may be sufficient to reduce AF susceptibility or maintenance, the human complexity might significantly limit promising pre-clinical findings [89,137,142,149] and request the concomitant target of several miRNAs.

More recently, a bioinformatic analysis of differentially expressed genes (DEGs), involved in AF and stroke-related complications, was performed [130] on a Gene Expression Omnibus (GEO) analysis of datasets GSE79768 and GSE58294, respectively. After data acquisition, an extensive target prediction and network analyses methods were carried out to assess protein–protein interaction (PPI) networks, Gene Ontology (GO) terms, pathway enrichment for DEGs, and co-expressed DEGs, coupled with corresponding predicted miRNAs, specifically involved in AF and related stroke. Multiple DEGs were identified. In particular, in left atrial specimens and cardioembolic stroke blood samples 489, 265, 518, and 592, deregulated genes were identified at < 3, 5, and 24 h, respectively. Different profiles were present in AF, stroke, and AF-related stroke. In fact, LRRK2, CALM1, CXCR4, TLR4, CTNNB1, and CXCR2 were found most in AF, while CD19, FGF9, SOX9, GNGT1, and NOG may be associated with stroke. Finally, in AF-related stroke, there were co-expressed DEGs of ZNF566, PDZK1IP1, ZFHX3, and PITX2, coupled with corresponding predicted miRNAs: **miR-27a-3p, miR-27b-3p,** and **miR-494-3p**.

Table 1 recapitulates the studies showing the presence of specific miRNAs as biomarkers in AF, their regulation, their gene target, and their clinical implications. 

### 4.2. miRNAs as Innovative Molecular Target Treatment in AF—Clinical Trials

To date, available results from clinical studies suggest that plasma levels of several miRNAs are associated with AF [153]. Unfortunately, the majority of these investigations were limited to cross-sectional analysis, or analyzed subgroups of AF patients, or were hypothesis-driven and restricted to subsets of specific miRNAs. Thus, as explained previously, current data are not sufficient to propose specific miRNAs as robust biomarkers, or even less, as target treatments for AF patients.

On 27 February 2023, a search with the key terms “atrial fibrillation and miRNAs” from the clinicaltrials.gov website resulted in 7 registered studies. At the moment, the majority are currently recruiting (NCT04710745; NCT05179902; NCT05743829) or have an unknown status (NCT04134793; NCT03855540); one has been completed but results are still not published (NCT02937077); the last one has been completed and outcomes are available (NCT03130985). 

In the clinical trial Correlation of Atrial Fibrillation Recurrence After Bipolar Radiofrequency Ablation With microRNA Expression—NCT04134793, miR-1, miR-19, miR-23, and miR-409 are enlisted among miRNAs whose expression levels changed in AF patients, who underwent catheter ablation, compared with normal sinus patients. This trial has been registered in 2019 and, at present, is not yet recruiting.

Another trial registered in 2019, Atrial Fibrillation in Relationship to Sleep Quality and Plasma Biomarkers (AFISBIO)-NCT03855540, has been designed to study the levels of plasmatic biomarkers in a high-risk cohort of patients with AF and several cardiovascular co-morbidities. Interestingly, the study aims to investigate the relation between sleep disorders and the AF episodes, based on the hypothesis that micro awakenings during the night increases sympathetic activity and may worsen heart rate variability. miR-1b, miR-133, miR-29b, miR-208a, miR-208b, miR-499, miR-19, miR-21, miR-124, miR-150, and miR-328 are among the biomarkers enlisted in the study protocol.

Published results from the clinical trial Tissue, Blood and Biomarkers to Predict Future Atrial Fibrillation (PREDICT-AF)—NCT03130985 (retrospectively registered on April 2017) demonstrates that atrial remodeling occurs long before incident AF and implies future potential for early patient identification and therapies to prevent AF. Unfortunately, among the circulating biomarkers investigated, no miRNAs levels have been reported [154].

The general objective of the study Atrial Fibrillation in Relationship to Plasma Biomarkers (AFISBIO II)—NCT04710745 is to assess the predictive ability of novel plasmatic biomarkers (especially apelin and miRNAs) on prevalent/incident AF in patients with high risk for AF and stroke, and to validate predictive models from previous studies based on co-morbidities, age, sex, BMI, NT-proBNP, FGF-23, IGF-1, and IGFBP-1 on prevalent/incident AF in patients with high risk for AF and stroke. Currently, the study is recruiting patients. No further details are available on the specific miRNAs to be assessed.

Likewise, the clinical trial Predictive Properties of Myocardial Fibrosis Biomarkers on the Outcome of Atrial Fibrillation Ablation (PROFIB-AF)—NCT05179902 aims to identify, among biological markers (ICTP, PICP, PIIINP, sRAGE, AGE, Galectin 3, sSt2, microRNAs) of myocardial fibrosis, those which can predict the recurrence risk after AF ablation. The study is recruiting patients starting from 2022. No further details are available on the specific miRNAs to be assessed.

The Atrial Fibrillation in Cryptogenic Stroke and TIA (NOR-FIB)—NCT02937077 study was planned to provide information on biomarkers that may be used to select cryptogenic TIA and stroke patients for long-term monitoring, as well as for information on the significance of short-term AF and optimal duration of cardiac rhythm monitoring. The study has been completed but no results have been published so far. Hence, the miRNAs evaluated, and their potential role, is not available at this time.

The Atrial Fibrillation Driver Study—NCT05743829 has been registered on clinical.trials.gov on 24 February 2023. The study is based on the concept that AF is associated with an increase in the scarring of the heart, and that, in turn, the scarring might be one of the drivers for abnormal heart rhythm. Investigation on the pattern of scarring is expected to gain a greater insight into the mechanisms of AF. Genome sequencing, RNA, and miRNAs sequencing (no further details) are enlisted among tools to examine the composition of the scarring tissue.

## 5. Conclusions

miRNAs are involved in every biological pathway through regulating expression of target genes/transcripts. miRNAs share many of the essential characteristics of a suitable biomarker: non-invasive measurability; a high degree of sensitivity and specificity, allowing early detection of pathological states; time-related changes during disease; a long half-life within the sample; and rapid and cost-effective laboratory detection. However, when assessing circulating miRNAs as biomarker of treatment efficacy, it must be considered that several drugs could affect their quantification and modify the results in blood samples. In the AF pathogenesis, a single miRNA may impact the expression of different targets (i.e., genes, proteins, cellular pathways), and several miRNAs may modulate a single target. Consequently, reproducing the in vivo conditions, and excluding the interactions of several miRNAs on a single selected target, is still a significant challenge.

One of the most debated aspects on the role of miRNAs is whether they act as “simple” biomarkers or causative mediators in AF. Although, in animal models, the ability of individual miRNAs to modulate cardiac phenotypes suggests that regulated expression of miRNAs may be a cause, rather than simply a consequence of cardiac remodeling [155,156], their role in AF has not been conclusively clarified. In human investigations, characterized by additional complexity, this is still an unresolved issue; currently available studies on the role of miRNAs in AF are diverse, with various designs and focuses, and often show contradicting results. Thus, several unclear aspects must be addressed to allow full applicability of novel strategies based on miRNAs.

Nonetheless, miRNA’s involvement in the modulation of cellular proliferation and differentiation, ED, and cardiac myocyte function encourages research to identify miRNAs as new key players and as potential therapeutic approaches for the treatment of several cardiovascular diseases, including AF.

## Figures and Tables

**Figure 1 ijms-24-05307-f001:**
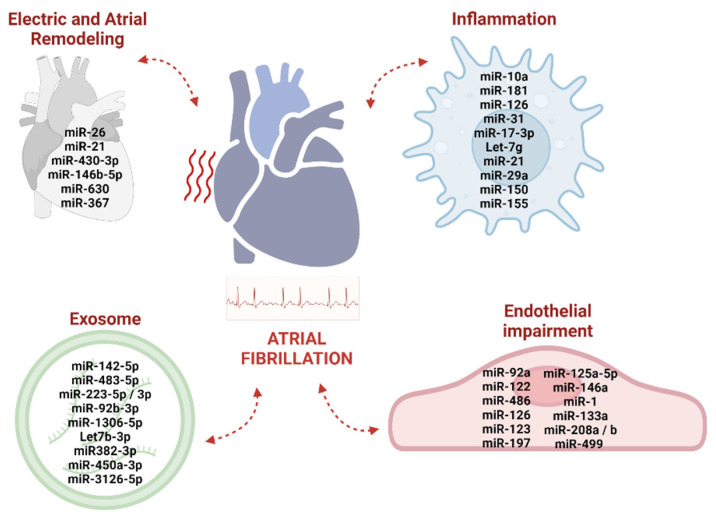
Schematic diagram highlighting the miRNAs most frequently associated with atrial fibrosis, vascular inflammation, and endothelial dysfunction, potentially contributing to AF onset and progression (description in the text).

**Table 1 ijms-24-05307-t001:** miRNAs as biomarkers in AF, their up and down regulation, their gene target, and their clinical implications.

miRNA (miR-)	UP/DOWN REGULATION	TARGET GENE	CLINICAL IMPLICATIONS	REF.
miR-26	UP DOWN	Kir2, TRPC3 channels L-type Ca^2+^	_	[35]
miR-21	UP DOWN	SPRY1, ERK/MAPK Smad7, TGF-β1, SPOUTY1, PTEN, PDCD4, RECK, TIP3	thrombogenesis and ionotropic functions, inflammation, fibrosis, hormones modulation	[90,91,92,151]
miR-29	DOWN	TGF-β	thrombogenesis, ionotropic functions, inflammation, fibrosis, hormones modulation	[96,151]
miR-430-3p, miR-146b-5p, miR-630, miR-367	UP/DOWN	mTOR, Hippo	_	[97]
miR-34a	UP	TASK1	_	[98]
miR-31	DOWN	VCAM-1, E-selectin, NF-kB, MCP1, IL-6, IL-8	inflammation, fibrosis, hormones modulation	[9,151]
miR-10a, miR-181, miR17-3p	DOWN	VCAM-1, E-selectin, NF-kB, MCP1, IL-6, IL-8	_	[9]
miR-126	DOWN UP	VCAM-1, HIF- 1TNF-α	_	[101,102]
miR-155	UP	eNos, TNF-α	_	[105]
miR-92a, miR-122, miR-486	UP/DOWN	Bcl-x, Caspase-3, ERK1/2	_	[107]
miR-223	UP	NLRP3	_	[112]
miR-197	UP	IL-6, CAM-1, VEGFA-VEGFR2	_	[113]
miR-125a-5p, miR-146a,	UP/DOWN	PTEN/Akt, SERCA2, NCX, VEGFR2, FGFR1	_	[114,115]
miR-1, miR-133a, miR-208a, miR-208b, miR-499	UP/DOWN	PTEN/Akt, SERCA2, NCX, VEGFR2, FGFR1, VSMCs, IGF1R	thrombogenesis and ionotropic functions, inflammation, fibrosis, hormones modulation	[114,115,151]
miR-27a-3p, miR-27b-3p, miR-494-3p	UP/DOWN	ZNF566, PDZK1IP1, ZFHX3, PITX2	_	[130]
miR-328	UP	TGF-β	fibrosis, hormones modulation	[151,152]

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
