# Peer review of "microRNAs as Biomarkers of Endothelial Dysfunction and Therapeutic Target in the Pathogenesis of Atrial Fibrillation"

_ijms, 2023, doi:10.3390/ijms24065307_

Round 1
Reviewer 1 Report
The review paper encompasses the recent literature on microRNAs involved in atrial fibrillation and atrial fibrosis pathogenesis. The manuscript is well organised and clear, with minor spelling and punctuation errors to be checked throughout.
My main comment is on the paragraph number 4. It is not clear which miRNA could indeed function as biomarkers to establish disease progression and drive the clinical decision making process on therapy and outcome assessment. The information is partially there but it should be, as the other paragraphs, made clearer, with a section dedicated only to miRNAs used as biomarkers for the listed disease conditions.
As the review focuses on the role of miRNAs as biomarkers and possible targets for therapy I would also add a section on clinical trials (closed and ongoing) regarding these aspects (microRNAs and Atrial fibrillation).
Finally, in order to make clear which microRNAs would be potential biomarkers I suggest adding the a column with this information within the Table 1, which at the moment only lists the miRNAs molecular function through their target.
Author Response
POINT-BY-POINT ANSWERS TO REVIEWERS’ COMMENTS
REVIEWER #1
The review paper encompasses the recent literature on microRNAs involved in atrial fibrillation and atrial fibrosis pathogenesis.
Q1. The manuscript is well organised and clear, with minor spelling and punctuation errors to be checked throughout. My main comment is on the paragraph number 4. It is not clear which miRNA could indeed function as biomarkers to establish disease progression and drive the clinical decision making process on therapy and outcome assessment. The information is partially there but it should be, as the other paragraphs, made clearer, with a section dedicated only to miRNAs used as biomarkers for the listed disease conditions.
A1. We are pleased that Reviewer 1 found our manuscript to be “well organized and clear” and thank her/him for useful comments and suggestions.
With respect to the specific miRNAs identified as biomarkers for disease progression and treatment decision, we agree with Reviewer’s comment to make this concept clear: several miRNAs have been identified as potentially involved in the pathogenesis of AF, with a variety of modulating effects often interconnected and overlapping. It is currently unknown which miRNAs may stand for a specific biomarker for disease progression, and therefore represent an effective tool to guide in clinical decision-making about therapy. Indeed, only few and very recent clinical trials are underway to address this point.
The most relevant miRNAs associated with Afib, inflammation and endothelial activities and potentially involved in AF are depicted in Figure 1 and discussed in the paragraph 2.
To clarify this point, the title of paragraph 4 has been modified as follows (pag. 12, line 450): “Could miRNA modulation become a biomarker guide in AF therapy?” and a sentence has been introduced (pag. 14, lines 505-509): “In addition, many clinical conditions may influence miRNAs expression, making particularly difficult, at this time, to establish which miRNA may display a decision-making role in the development of disease. Hopefully, among the many miRNAs candidates as biomarkers, the ongoing research will help to clarify their specific activities in modulating fundamental processes underlying AF.”
All text has been revised in order to correct minor spelling and punctuation errors.
Q2. As the review focuses on the role of miRNAs as biomarkers and possible targets for therapy I would also add a section on clinical trials (closed and ongoing) regarding these aspects (microRNAs and Atrial fibrillation).
A2. According to Reviewer’s recommendations the paragraph 4 has been implemented, with an additional section discussing the current clinical trials investigating miRNAs in atrial fibrillation (pag. 12-13, lines 547-605):
4.2 miRNAs as innovative molecular target treatment in AF - Clinical trials.
To date, available results from clinical studies suggest that plasma levels of several miRNAs are associated with AF [153]. Unfortunately, the majority of these investigations were limited to cross-sectional analysis, or analyzed subgroups of AF patients, or were hypothesis-driven and restricted to subsets of specific miRNAs. Thus, as explained previously, current data are not sufficient to propose specific miRNAs as robust biomarkers, or even less, as target treatments for AF patients.
On February 27, 2023, a search with the key terms "atrial fibrillation and miRNAs" from the clinicaltrials.gov website resulted in 7 registered studies. At the moment, the majority are currently recruiting (NCT04710745; NCT05179902; NCT05743829) or have an unknown status (NCT04134793; NCT03855540); one has been completed but results still not published (NCT02937077); the last one has been completed and outcomes are available (NCT03130985).
In the clinical trial Correlation of Atrial Fibrillation Recurrence After Bipolar Radiofrequency Ablation With microRNA Expression-NCT04134793, miR-1, miR-19, miR-23, miR-409 are enlisted among miRNAs whose expression levels changed in AF patients, who underwent catheter ablation, compared with normal sinus patients. This trial has been registered in 2019 and at present is not yet recruiting.
Another trial registered in 2019, Atrial Fibrillation in Relationship to Sleep Quality and Plasma Biomarkers (AFISBIO)- NCT03855540, has been designed to study the levels of plasmatic biomarkers in high-risk cohort of patients with AF and several cardiovascular co-morbidities. Interestingly, the study aims to investigate the relation between sleep disorders and the AF episodes, based on the hypothesis that micro awakenings during the night increase sympathetic activity and may worsen the heart rate variability. miR-1b, miR-133, miR-29b, miR-208a, miR-208b, miR-499, miR-19, miR-21, miR-124, miR-150 and miR-328 are among the biomarkers enlisted in the study protocol.
Published results from the clinical trial Tissue, Blood and Biomarkers to Predict Future Atrial Fibrillation (PREDICT-AF) - NCT03130985 (retrospectively registered on April 2017), demonstrates that atrial remodeling occurs long before incident AF and implies future potential for early patient identification and therapies to prevent AF. Unfortunately, among circulating biomarkers investigated, no miRNAs levels have been reported [154].
The general objective of the study Atrial Fibrillation in Relationship to Plasma Biomarkers (AFISBIO II) NCT04710745 is to assess predictive ability of novel plasmatic biomarkers (especially apelin and miRNAs) on prevalent/incident AF in patients with high risk for AF and stroke, and to validate predictive models from previous studies based on co-morbidities, age, sex, BMI, NT-proBNP, FGF-23, IGF-1 and IGFBP-1 on prevalent/incident AF in patients with high risk for AF and stroke. Currently the study is recruiting patients. No further details are available on the specific miRNAs to be assessed.
Likewise, the clinical trial Predictive Properties of Myocardial Fibrosis Biomarkers on the Outcome of Atrial Fibrillation Ablation (PROFIB-AF) – NCT05179902 aims to identify, among biological markers (ICTP, PICP, PIIINP, sRAGE, AGE, Galectin 3, sSt2, microRNAs) of myocardial fibrosis, those which can predict the recurrence risk after AF ablation. The study is recruiting patients starting from 2022. No further details are available on the specific miRNAs to be assessed.
The Atrial Fibrillation in Cryptogenic Stroke and TIA (NOR-FIB) NCT02937077 study was planned to provide information on biomarkers that may be used to select cryptogenic TIA and stroke patients for long-term monitoring, as well as information on the significance of short-term AF and optimal duration of cardiac rhythm monitoring. The study has been completed but no results have been published so far. Hence, the miRNAs evaluated and their potential role of is not available at this time.
The Atrial Fibrillation Driver Study - NCT05743829 has been registered on clinical.trials.gov on feb 24, 2023. The study is based on the concept that AF is associated with an increase in the scarring of the heart, and in turn the scarring might be one of the drivers for abnormal heart rhythm. Investigation on the pattern of scarring is expected to gain a greater insight into the mechanisms of AF. Genome sequencing, RNA and miRNAs sequencing (no further details) are enlisted among tools to examine the composition of the scarring tissue.
References have been updated accordingly.
Q3. Finally, in order to make clear which microRNAs would be potential biomarkers I suggest adding a column with this information within the Table 1, which at the moment only lists the miRNAs molecular function through their target.
A3. To satisfy the Reviewer's request, Table 1 has been modified, with a new column related to clinical implications for each enlisted miRNAs as possible targets for therapy. The text has been modified accordingly (pag. 15, lines 540-545): “Table 1 recapitulates the studies showing the presence of specific miRNAs as biomarkers in AF, their regulation, their gene targets, and their clinical implications.
Table 1. miRNAs involved as biomarkers in AF, their up and down regulation, their gene targets, and their clinical implications.”
References have been updated accordingly.

Reviewer 2 Report
The present review by Desantis et al. aims to provide a current understanding of the mechanisms controlling AF onset and progression. They summarize the most recent features of AF associated with miRNAs and relate them to potentially underlying mechanisms. Overall, this review is relatively comprehensive and logical. I only have a minor comment as follows.
1 It is known that sympathetic stress can trigger AF. What is the associate mechanism by which sympathetic adrenergic receptor activation induces AF?
2 The authors highlight the miRNAs most frequently associated with atrial fibrosis, vascular inflammation, and endothelial dysfunction, potentially contributing to AF onset and progression. What pathologic stimulus changes these miRNAs? Is it AF itself or other pathologic stimuli? It will be nice to discuss this
Author Response
POINT-BY-POINT ANSWERS TO REVIEWERS’ COMMENTS
REVIEWER #2
The present review by Desantis et al. aims to provide a current understanding of the mechanisms controlling AF onset and progression. They summarize the most recent features of AF associated with miRNAs and relate them to potentially underlying mechanisms. Overall, this review is relatively comprehensive and logical. I only have a minor comment as follows.
We are grateful to Reviewer 2 for his/her positive evaluation of our manuscript and valuable comments and suggestions.
Q1. It is known that sympathetic stress can trigger AF. What is the associate mechanism by which sympathetic adrenergic receptor activation induces AF?
A1. To comply with the Reviewer’s request, a new subparagraph has been included in paragraph 2.1 describing the influence and potential AF-triggering effects of sympathetic receptor activation (pag. 4, lines 131-151):
Sympathetic receptor activation - Autonomic imbalance is one of the most important pro-arrhythmic modulating factor, favoring both atrial ectopic activity and re entry. Sympathetic stimulation of adrenergic receptors enhances all processes controlling Ca2+ entry, storage and release in cardiomyocytes, promoting significant and heterogeneous changes of atrial electrophysiology that may represent both a substrate and a trigger for the onset of AF.
Downstream β1-adrenergic receptors, activation of PKA-dependent phosphorylation of several Ca2+-handling proteins and cyclic adenosine monophosphate (cAMP) production [27] results in increased sarcoplasmic-reticulum (SR) Ca2+load. This effect, together with hyperphosphorylation of cardiac ryanodine-receptor channel type 2 (RYR2), can cause diastolic SR Ca2+ leak, promoting delayed afterdepolarization (DAD), which is a recognized source of ectopic activity in the onset and maintenance of AF [27-28].
In addition to β1-adrenoreceptors, stimulation of α1-receptor-mediated signaling may also contribute to the SR Ca2+leak in atrial cardiomyocytes, as well as to the maintenance of AF [29]. Activation of α1-adrenergic receptors can also inhibit inward rectifying K+ current (IK), which is important in setting the resting potential and the repolarization reserve. By inhibiting IK, α1-adrenergic activation may enhance stimulation of cardiac neurons and increase automaticity, promoting AF onset [15, 29]. Thus, sympathetic receptor activation may induce atrial ectopic activity via multiple mechanisms, contributing to the pathogenesis of atrial arrhythmias which in turn exacerbate atrial autonomic imbalance.
Q2. The authors highlight the miRNAs most frequently associated with atrial fibrosis, vascular inflammation, and endothelial dysfunction, potentially contributing to AF onset and progression. What pathologic stimulus changes these miRNAs? Is it AF itself or other pathologic stimuli? It will be nice to discuss this.
A2. This reviewer’s question highlights maybe the most debated aspect on the role of miRNAs in AF, whether as “simple” biomarkers or causative mediators. In general, miRNAs are involved in every biological pathway through regulating expression of target genes/transcripts Currently available studies investigating the role of miRNAs in AF are diverse, with various designs and focuses, and show contradicting results. The exact pathology behind the associations of many miRNAs with cardiovascular diseases is not completely understood. Although in animal models the ability of individual miRNAs to modulate cardiac phenotypes suggests that regulated expression of miRNAs is a cause rather than simply a consequence of cardiac remodeling, conclusive evidence in human investigation is not possible at this point.
These concepts have been discussed and included in the paragraph of the conclusions: (pag. 18, lines 607-608) “miRNAs are involved in every biological pathway through regulating expression of target genes/transcripts.” and (pag. 19, lines 619-627) “One of the most debated aspects on the role of miRNAs is whether they act as “simple” biomarkers or causative mediators in AF. Although in animal models the ability of individual miRNAs to modulate cardiac phenotypes suggests that regulated expression of miRNAs may be a cause, rather than simply a consequence of cardiac remodeling, their role in AF has not been conclusively clarified. In human investigations, characterized by additional complexity, this is still an unresolved issue: currently available studies on the role of miRNAs in AF are diverse, with various designs and focuses, and often show contradicting results. Thus, several unclear aspects must be addressed to allow full applicability of novel strategies based on miRNAs.”

Round 2
Reviewer 1 Report
thank you for adding the informations. I think now the review will catch many readers!
Author Response
We are grateful to the Reviewer 1 for her/him positive evaluation.